# Quantitative Assessment of Balance Function Characteristics in Older Patients with Orthostatic Hypotension

**DOI:** 10.3390/geriatrics8050103

**Published:** 2023-10-18

**Authors:** Yao Cui, Bo Liu, Jian Zhou, Qian Liu, Hui Ye

**Affiliations:** 1Department of Geriatrics, Beijing Tongren Hospital, Capital Medical University, Beijing 100730, China; xiaoniao912@mail.ccmu.edu.cn (Y.C.); zhoujian@mail.ccmu.edu.cn (J.Z.); liuqian@mail.ccmu.edu.cn (Q.L.); yehui@mail.ccmu.edu.cn (H.Y.); 2Department of Otolaryngology Head and Neck Surgery, Beijing Tongren Hospital, Capital Medical University, Beijing Institute of Otolaryngology, Key Laboratory of Otolaryngology Head and Neck Surgery (Capital Medical University), Ministry of Education, Beijing 100730, China

**Keywords:** older, balance function, sensory organization test, orthostatic hypotension

## Abstract

Background: Orthostatic hypotension (OH) is a common blood pressure abnormality in older adults that makes them prone to balance disorders and falls. The maintenance of balance relies on a complex regulatory system. The use of computerized dynamic posturography (CDP) can provide a quantitative evaluation of balance function. The objective of this study was to utilize CDP to measure balance indicators in older individuals with OH. Methods: A total of 303 older adults were divided into an OH group of 91 and a non-OH group of 212. Various factors, including chronic diseases, medication history, laboratory tests, and balance indicators, were compared between the two groups. Results: ① Participants with OH had more chronic diseases, including coronary heart disease (*p* = 0.049) and a history of falls (*p* < 0.001), than those without OH. A history of multiple medications in the OH group was significantly more likely than in the non-OH group (*p* = 0.006). ② There was a significant reduction in the composite equilibrium score (SOT-COM) (*p* < 0.001), vision ratio score (VIS) (*p* < 0.001), vestibular ratio score (VEST) (*p* < 0.001), and directional control (DCL) (*p* = 0.028) in the OH group. ③ A logistic regression analysis revealed that SOT-COM was a significant independent factor associated with OH. The area under the curve (AUC) of SOT-COM was 0.833 (95% confidence interval: 0.778–0.887, *p* < 0.001), with a sensitivity of 0.826 and a specificity of 0.759. Conclusions: This study demonstrates that older individuals with OH are more prone to falls, due to decreased sensory integration for balance.

## 1. Introduction

With the population aging at an increasing rate, the health concerns of older adults are gaining more attention. Orthostatic hypotension (OH) is a common abnormality in blood pressure [1]. Normally, people are expected to recover their blood pressure to pre-standing levels within 30–40 s after standing up. However, if the recovery of blood pressure is slow, OH may occur [2]. OH is a condition characterized by a drop in systolic blood pressure of at least 20 mmHg or a decrease in diastolic blood pressure of at least 10 mmHg within 3 min of assuming an upright position [3,4].

As individuals age, their ability to regulate blood pressure decreases. This often leads to a significant increase in the occurrence of OH. OH affects one-fifth of older individuals living in the community [2]. When patients with OH stand up or sit up, their blood pressure drops, which can lead to inadequate blood flow to the brain. This can cause symptoms such as dizziness, vertigo, and syncope. In severe cases, dangerous events, such as falls, may occur. In older individuals, there is a decline in sympathetic nervous system activity and a rise in parasympathetic nervous system activity, which can make them more susceptible to balance disorders [5]. In addition, there are several factors that affect balance in older individuals, including reduced muscle strength and visual and hearing impairments. These factors can increase the likelihood of falls among older adults. In other words, older patients with OH are highly susceptible to balance disorders and falls.

The ability to maintain balance depends on visual, vestibular, and proprioceptive inputs, which are integrated by the brain to generate motion and represent the body’s position [6]. A systematic review examining the relationship between OH and physical function revealed that only a small number of studies indicated a correlation between OH and decline in physical function. Specifically, 7 out of 14 studies reported a decrease in balance abilities and 1 out of 5 studies observed gait abnormalities, but no correlations between OH and physical frailty, exercise endurance, and level of physical activity were found [3]. A study conducted on older individuals who had resided in nursing homes for a prolonged period revealed that OH was not directly associated with falls, but there was a significant risk of experiencing recurrent falls due to OH [7]. Other studies found that dynamic physical performance tests, such as the 4 m walk test, chair stand test, and timed up-and-go test, were significantly associated with OH [8]. Some studies found a significant association between asymptomatic OH and unexplained falls. Prospective studies have also suggested that OH is an independent predictor of falls [9].

These inconsistent results may be attributed to variations in the comprehension and implementation skills of both the evaluators and participants, resulting in inconsistencies in clinical practice. Moreover, most existing studies have examined the correlation between OH and aspects such as gait, motor ability, and physical function. However, maintaining balance also requires the participation of sensory input and integration ability. Therefore, more accurate and objective evaluations can be achieved through the use of quantitative assessment techniques. Computerized dynamic posturography (CDP) is a valuable tool for measuring both static and dynamic balance functions, and it includes sensory organization testing (SOT) and limits-of-stability testing (LOS) [10]. SOT assesses various sensory inputs, including proprioception, vision, and vestibular sensation. On the other hand, LOS evaluates the motor output performance, including reaction time (RT), maximum deviation of the pressure center (MXE), and directional control (DCL). These measures reflect the stability limits of the human body.

This study aimed to utilize CDP to quantitatively measure balance indicators in older individuals with OH, in order to gain a better understanding of the balance characteristics in this population.

## 2. Methods

### 2.1. Participants

Between January 2022 and January 2023, we enrolled a total of 303 older patients from an outpatient department for geriatric medicine, including 217 males and 86 females, who agreed to participate in this study. The patients were examined by geriatric specialists who received professional training. The patients’ right upper limb blood pressure was measured in a supine position for 15 min and then 1 and 3 min after changing from supine to an upright position. A decrease in systolic blood pressure of ≥20 mmHg and/or a decrease in diastolic blood pressure of ≥10 mmHg after 1 or 3 min in the upright position, compared to the supine position, were considered to indicate the presence of OH. Based on the diagnostic criteria for OH, the patients were classified into two groups: 91 patients in the OH group and the remaining 212 patients in the non-OH group.

The inclusion criteria were as follows: ① age ≥ 60 years old, ② capable of walking independently for a distance of 30 m, ③ possessing the ability to understand and communicate with others, ④ consenting to participate in the study and undergo relevant tests, and ⑤ had not smoked or drunk alcohol for at least 1 year.

The exclusion criteria were as follows: ① inability to understand and accurately cooperate with the examination, ② currently experiencing an acute heart attack, atrial fibrillation, or lung or kidney disease, ③ having severe cerebrovascular or central nervous system disease (such as non-vascular-induced white matter lesions, tumors, stroke, Parkinson’s disease, dementia with Lewy bodies, or traumatic brain injury), ④ suffering from acute inflammatory infection, ⑤ having conditions that affect hemodynamics, such as diarrhea or insufficient food intake, and ⑥ long-term use of α-receptor blockers, long-term use of antidepressants, or use of sedative or hypnotic drugs in the past 2 weeks.

### 2.2. Data Collection

The participants’ age, sex, height, and weight were measured, and their body mass index (BMI) was calculated as weight (kg)/height^2^ (m^2^). Diagnoses and comorbidities were obtained from medical records. Polypharmacy was defined as the use of five or more medications.

### 2.3. Laboratory Tests

A fasting sample of 10 mL of venous blood was collected from the elbow between 8:00 a.m. and 10:00 a.m. and promptly sent to the laboratory. Hemoglobin (Hgb), albumin (ALB), blood urea nitrogen (BUN), serum creatinine (Cr), uric acid (UA), triglycerides (TGs), total cholesterol (TC), low-density lipoprotein cholesterol (LDL-C), high-density lipoprotein cholesterol (HDL-C), glycosylated hemoglobin A1c (HbA1c), and 25-hydroxyvitamin D (25-OH-D) were all measured. The estimated glomerular filtration rate (eGFR) was calculated using the CKD-EPI formula. Additionally, the arterial stiffness index (AI) was calculated as (TC-HDL-C)/HDL-C.

### 2.4. Computer Dynamic Posturography (CDP)

The EquiTest Dynamic Balance Bench Tester and Balance Test Board (NeuroCom International, Clackamas, OR, USA) was used to assess the balance function (Figure 1). Subjects were instructed to respond to the given conditions, and the data were analyzed and processed with a computer. The SOT and LOS results were automatically calculated and completed.

#### 2.4.1. Sensory Organization Test (SOT)

The SOT included the estimation of the patient’s center-of-gravity displacements in six different sensorial information conditions. Each of the six conditions (lasting for 20 s) was repeated three times [11] as follows:Fixed surface, fixed visual surroundings, eyes open.Fixed surface, eyes closed.Fixed surface, moving visual surroundings, eyes open.Moving surface, fixed visual surroundings, eyes open.Moving surface, eyes closed.Moving surface, moving visual surroundings, eyes open.

The somatosensory ratio score (SOM), vision ratio score (VIS), and vestibular ratio score (VEST) were calculated according to the scores under different conditions. SOM: (mean score of condition 2/mean score of condition 1) × 100. VIS: (mean score of condition 4/mean score of condition 1) × 100. VEST: (mean score of condition 5/mean score of condition 1) × 100.

The composite equilibrium score (SOT-COM) was a value of the arithmetic mean of the percentage of balance obtained in each SOT condition, and it reflected the ability to integrate balanced sensory input. A score of 100 indicated a small swing in the body’s center of gravity, while a score of 0 indicated a fall or approaching the limit of balance.

#### 2.4.2. Limits-of-Stability Test (LOS)

The participants stood on a platform and faced forward while stabilizing their center of gravity at the center of the testing area. When a signal appeared on the screen, they quickly moved their body’s center of gravity to the target area and maintained stability. After 10 s, they moved their center of gravity back to the center of the testing area. The participants needed to reach eight points (front, back, left, right, left-front, right-front, left-back, and right-back) around themselves to obtain comprehensive results through a computer analysis and processing [12].

We primarily observed the following three indicators: ① RT was the duration (in seconds) between the start of a trial and the moment when the participant began to move towards the intended target. ② MXE was the maximum displacement of the pressure center observed during the entire experimental process, expressed as a percentage of the maximum stability limit. ③ DCL refers to the ability of the participant to control the direction of their movement toward the target. DCL refers to the degree to which the intended movement was accurately carried out, and was measured as the percentage of straightness in the displacement trajectory of the center of pressure.

## 3. Statistical Analyses

The statistical analysis was conducted using SPSS (Version 22). Quantitative data that were normally distributed were expressed as the mean and standard deviation (mean ± SD), while non-normally distributed data were expressed as quartiles (P25–P75). The independent-samples *t*-test and the Mann–Whitney U test were utilized to compare group differences. Qualitative data are typically presented as a proportion or percentage. A Pearson Chi-squared (χ^2^) test was used to compare the differences between groups. Binary logistic regression was conducted to identify the factors influencing OH. The analysis yielded Exp (B) and its corresponding 95% confidence intervals (CIs). A receiver operating characteristic (ROC) curve was drawn to analyze the predictive value of the SOT-COM on OH. A *p*-value of less than 0.05 was considered statistically significant.

## 4. Results

Table 1 shows the characteristics of the study participants. Of 303 older adults, 91 were in the OH group and 212 were in the non-OH group. No significant differences in age, sex, or BMI (*p* > 0.05) were observed between the groups, and thus, the two groups were essentially matched. Participants with OH had more chronic diseases, including coronary heart disease (*p* = 0.049) and a history of falls (*p* < 0.001), than those without OH. A history of multiple medications in the OH group was significantly higher than that in the non-OH group (*p* = 0.006).Table 2 shows that there were no statistically significant differences between the two groups in terms of laboratory indicators.As shown in Table 3, there was a significant decrease in SOT-COM (*p* < 0.001), VIS (*p* < 0.001), VEST (*p* < 0.001), and DCL (*p* = 0.028) in the OH group, compared to the non-OH group, but there were no significant differences in SOM, RT, or MXE between the two groups (*p* > 0.05).A regression analysis was conducted in this study, using various balance indicators as independent variables to analyze the incidence of OH as a binary variable. After adjusting for age, gender, and body mass index, only SOT-COM was found to have a negative correlation with the incidence of OH (OR = 0.904; 95% CI: 0.882–0.927; *p* < 0.001). Further adjustments for factors such as history of falls, coronary heart disease, history of multiple medications, and laboratory tests revealed that only SOT-COM was negatively correlated with the incidence of OH (OR = 0.884; 95% CI: 0.832–0.940; *p* < 0.001).An ROC curve was plotted for all participants, using the occurrence of OH as the positive rate, and the area under the curve was calculated (Figure 2). The AUC of SOT was 0.833 (95% confidence interval: 0.778–0.887; *p* < 0.001), with a sensitivity of 0.826 and a specificity of 0.759.

## 5. Discussion

Older individuals often experience physiological changes, including those affecting the cardiovascular, nervous, muscular, and skeletal systems, which can make them more susceptible to OH. This study found that the prevalence of hypertension, coronary heart disease, diabetes, cerebrovascular disease, and chronic kidney disease in the OH group was higher than that in the non-OH group, which is consistent with the findings of previous studies [13,14]. This finding holds true, even though there were no statistically significant differences in age and gender between the two groups. Orally administered α-receptor blockers, sedative–hypnotics, antipsychotics, and antidepressant drugs all have the potential to cause OH [15]. Therefore, this study excluded patients who were taking these drugs. However, it was still observed that the proportion of OH patients taking five or more drugs was significantly higher than that of older individuals without OH. This suggests that polypharmacy alone could increase the risk of OH.

This study found that the incidence of falls in older people with OH was significantly higher than that in the non-OH group, reaching a surprising 73.63%. The occurrence of falls is related to multiple factors, among which balance ability is a crucial factor. Vestibular perception is primarily responsible for detecting the rotation and directional acceleration of the human body. It gathers motion information from the head, which is used to establish a three-dimensional spatial positioning system that helps maintain the body’s balance and posture [16,17]. The function of vision is to sense external light stimuli, process visual information, perceive spatial information about the surrounding environment, and detect the connections, movements, and positional changes between different parts of the body, thereby assisting in maintaining balance control [18,19]. Proprioception is distributed throughout the body, including peripheral receptors, joint receptors, and skin receptors [20,21]. The nervous system collects, integrates, and processes this information before outputting it as motion commands to maintain the basic form and balance of the human body.

This study utilized computer technology to measure human balance and postural control, also known as computerized dynamic posturography (CDP) testing. CDP testing includes the sensory organization test (SOT) and the limits-of-stability test (LOS). The sensory organization test (SOT) assesses the vestibular (VEST), visual (VIS), and somatosensory (SOM) senses. It calculates a composite equilibrium score (SOT-COM) by weighting and averaging the scores from these three senses. Thus, the SOT-COM reflects the ability to integrate the senses of balance. The VIS, VEST, and SOT-COM scores of the OH group were significantly lower than those of the non-OH group. The visual and vestibular senses can directly impact balance function. The Survey of Health, Ageing and Retirement in Europe found that individuals with poorer vision—with or without health improvements—had a 19% and 33% higher risk of falls, respectively, compared with those with good vision and health [22]. The vestibular system is essential for maintaining balance in both static and dynamic conditions, as it provides critical information about spatial orientation. Approximately 50% of individuals aged 60 and above suffer from vestibular dysfunction [23]. The physiological degeneration of the vestibular system can be easily ignored by older individuals due to its slow progression. However, damage to the body’s balance ability accumulates over time, resulting in exercise intolerance, instability, and gait issues [24]. This study also found that the ability of older patients in the OH group to control their direction decreased. Falls are more likely to occur, especially during sudden turns [25]. SOT-COM represents the ability to integrate proprioception, vision, and vestibular perception. In this study, the OH group exhibited a decrease in sensory integration ability, as evidenced with a significant decrease in SOT-COM compared to the non-OH group.

LOS reflects a person’s ability to consciously move their body in any direction without falling or losing balance. RT reflects the subject’s reaction speed. MXE is the maximum distance the body can reach during the tilting process. DCL is a measure of the body’s ability to accurately regulate the direction of movement. Although there were no statistically significant differences between the two groups in these indicators, it was still evident that older patients with OH had longer reaction times and shorter maximum reachable distances. Furthermore, this study took other potential factors that could impact OH into account. For instance, diabetes can directly cause autonomic nervous dysfunction or affect capillary and renal vessels, leading to the development of OH [26]. However, the prevalence of diabetes and chronic kidney disease, as well as levels of HbA1C and eGFR, did not differ between the two groups in this study. Other studies have shown that arteriosclerosis is also a contributing factor to the occurrence of OH [27]. This study utilized blood lipid indicators to calculate the arteriosclerosis index (AI). The results showed no significant differences between the two groups. A decrease in BMI [28] and autonomic dysfunction caused by low levels of 25(OH)D3 [29] may also contribute to the occurrence of OH. However, these indicators did not show any statistically significant differences between the two groups in this study and, therefore, are not considered to be the main influencing factors for OH.

A further regression analysis revealed that only SOT-COM was an independent correlating factor. After adjusting for age, gender, body mass index, history of falls, and laboratory indicators, SOT-COM remained an independent correlating factor. In other words, balance disorders in older patients with OH primarily stem from a decline in their ability to integrate sensory information related to balance. The ROC curve indicates that SOT-COM has a strong predictive value for OH in older individuals.

## 6. Conclusions

This study found that older individuals with OH have a significantly higher incidence of falls. Quantifying and accurately assessing balance in older people with OH can effectively prevent falls. Older patients with OH experience a decline in their visual, vestibular, SOT-COM, and directional control abilities. The regression analysis indicated that SOT-COM is an independent factor that is associated with OH. The ROC indicated that SOT-COM has a high predictive value for OH in older individuals. This study suggests that balance disorders in older patients with OH primarily result from their decreased capacity to integrate sensory information associated with balance.

## Figures and Tables

**Figure 1 geriatrics-08-00103-f001:**
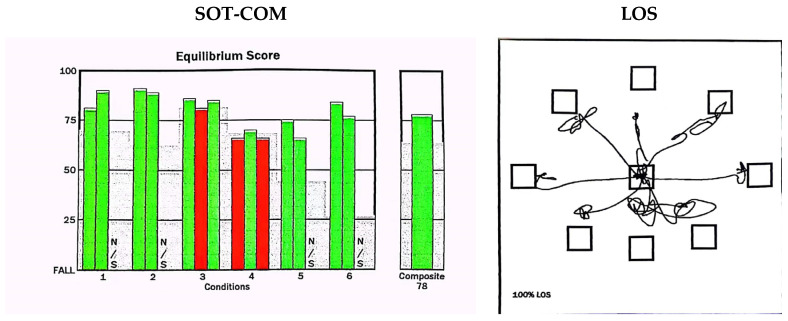
Schematic diagram of CDP. **SOT:** sensory organization test; SOT-COM: the composite equilibrium score was a value of the arithmetic mean of the percentage of balance obtained in each SOT condition. The vertical axis represents the fall score: a score of 100 indicated a small swing in the body’s center of gravity, while a score of 0 indicated a fall or approaching the limit of balance. Under 6 conditions, the computer automatically calculates SOM, VIS, and VEST scores, with green indicating normal, red indicating abnormal, and N/S indicating undetectable. **LOS:** limits-of-stability test—the screen displays signals in eight different directions, including front, back, left, right, left-front, right-front, left-rear, and right-rear.

**Figure 2 geriatrics-08-00103-f002:**
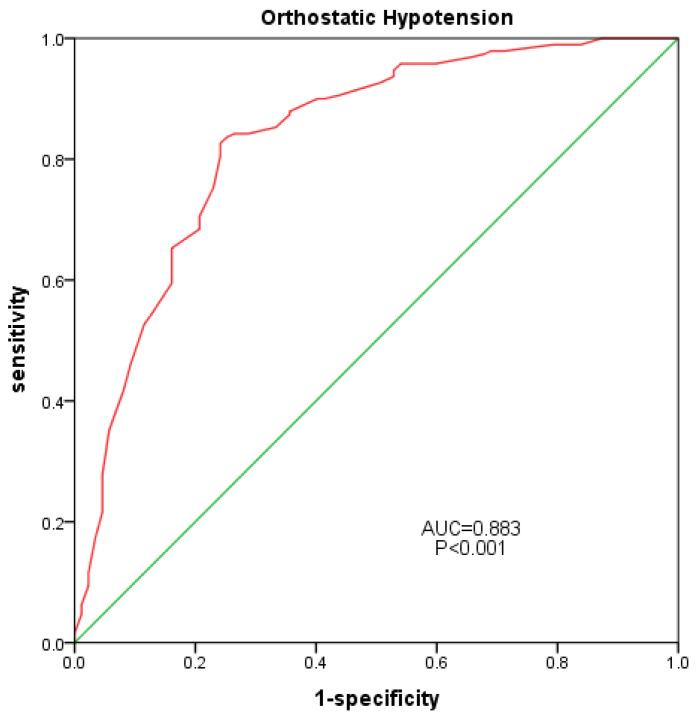
Correlation between SOT-COM and OH. Red line: SOT-COM, Green line: diagnostic reference line.

**Table 1 geriatrics-08-00103-t001:** Characteristics of the older adults in the OH and non-OH groups.

	OH Group (91 Cases)	Non-OH Group (212 Cases)	|X^2^/T/Z|	*p*
Age (years)	80.10 ± 8.12	79.04 ± 8.74	0.989	0.323
Sex (male, %)	67.03	73.58	1.345	0.246
BMI	24.02 ± 3.30	23.86 ± 2.87	0.439	0.661
Hypertension (Yes, %)	74.73	68.57	1.155	0.283
Coronary heart disease (Yes, %)	52.75	40.48	3.876	0.049 *
Diabetes (Yes, %)	47.25	38.10	2.203	0.138
Cerebrovascular disease (Yes, %)	23.08	18.57	0.808	0.369
Chronic kidney disease (Yes, %)	8.79	8.57	0.004	0.950
History of falls (Yes, %)	73.63	32.10	44.498	<0.001 *
Multiple medication history (≥5, %)	63.74	46.67	7.409	0.006 *

BMI: body mass index. * *p* < 0.05.

**Table 2 geriatrics-08-00103-t002:** Laboratory indicators of the older adults in the OH and non-OH groups.

	OH Group (91 Cases)	Non-OH Group (212 Cases)	|X^2^/T/Z|	*p*
Hgb (g/L)	132.15 ± 16.54	135.13 ± 13.32	1.430	0.154
ALB (g/L)	41.40 ± 6.61	40.56 ± 3.87	1.075	0.285
SCr (µmol/L)	93.83 ± 58.90	86.20 ± 32.02	1.113	0.268
BUN (mmol/L)	6.47 ± 2.35	6.13 ± 1.89	1.232	0.219
UA (µmol/L)	356.17 ± 98.83	349.41 ± 85.08	0.570	0.569
TG (mmol/L)	1.35 ± 0.98	1.36 ± 0.75	0.073	0.942
TC (mmol/L)	4.06 ± 0.93	4.26 ± 0.94	1.643	0.102
HDL-C (mmol/L)	1.29 ± 0.43	1.39 ± 0.44	1.684	0.093
LDL-C (mmol/L)	2.22 ± 0.81	2.33 ± 0.80	1.039	0.300
AI	2.41 ± 1.19	2.30 ± 1.03	0.800	0.425
HbA1C (%)	7.20 ± 1.59	6.90 ± 1.97	0.985	0.326
25(OH)D3 (ng/mL)	24.49 ± 12.43	25.58 ± 12.30	0.589	0.556
eGFR (mL/min/1.73 m^2^)	71.99 ± 20.20	73.18 ± 17.34	0.488	0.626

Hgb: hemoglobin, ALB: albumin, SCr: serum creatinine, BUN: blood urea nitrogen, UA: uric acid, TG: triglyceride, TC: total cholesterol, HDL-C: high-density lipoprotein cholesterol, LDL-C: low-density lipoprotein cholesterol, HbA1C: hemoglobin A1C, AI: arterial stiffness index, eGFR: estimated glomerular filtration rate.

**Table 3 geriatrics-08-00103-t003:** Comparison of balance indicators of the older adults in the OH and non-OH groups.

	OH Group (91 Cases)	Non-OH Group (212 Cases)	|X^2^/T/Z|	*p*
SOM (M (Q1, Q3))	98 (95, 100)	98 (95, 100)	1.205	0.228
VIS (M (Q1, Q3))	70 (52, 80)	80 (70, 85)	6.088	<0.001 *
VEST (M (Q1, Q3))	15 (5, 60)	65 (60, 80)	8.173	<0.001 *
SOT-COM (M (Q1, Q3))	53 (42, 66)	76 (69, 80)	8.898	<0.001 *
RT (s)	0.95 ± 0.22	0.90 ± 0.26	1.357	0.176
MXE (%)	66.60 ± 13.01	68.80 ± 11.45	1.386	0.167
DCL (%)	67.35 ± 9.41	70.26 ± 9.82	2.210	0.028 *

SOM: Somatosensory; VIS: Vision; Vest: Vestibular; SOT-COM: Composite Equilibrium Score; RT: Reaction Time; MXE: Maximum Excursion; DCL: Directional Control. * *p* < 0.05.

## Data Availability

The datasets used in the current study are not publicly available due to them containing information that could compromise research participant privacy but are available from the corresponding author on reasonable request.

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
