# Peer review of "Quantitative Assessment of Balance Function Characteristics in Older Patients with Orthostatic Hypotension"

_geriatrics, 2023, doi:10.3390/geriatrics8050103_

Round 1
Reviewer 1 Report
See attachment.

Author Response
On behalf of my co-authors, I would like to thank you for their positive and constructive comments and suggestions. Based on your modification suggestions, we make the following response.
- In the abstract I would just specify the number of OH vs non-OH patients. I believe this information is more relevant than the total number of participants.
Answer: Thanks for the suggestion, I added the number of people in each group to the summary
- Section 2.4: the CDP should be described in more detail. In particular there are no information about how SOM, VIS and VEST are computed. This is an important information
Answer: I have added the calculations for SOM, VIS, and VEST to section 2.4.1
- The quality of Figure 1 needs to be improved. Please use vector format to avoid loss of quality. Also, it is not clear what FALL on the vertical axis means in this context and why there are two red bars and the other are green.
Answer: I have modified Figure 1 and added annotations, hoping to meet your requirements. The vertical axis represents the fall score. A score of 100 indicated a small swing in the body's center of gravity, while a score of 0 indicated a fall or approaching the limit of balance. Under 6 conditions, the computer automatically calculates SOM, VIS, and VEST scores, with green indicating normal, red abnormal and N/S undetectable.
4.Table I. The caption says that values are presented as median (interquartile range) but it seems rather mean ± standard deviation.
Answer: Yes, I made an error, it should be "mean ± standard deviation “.
- The quality of Figure 2 should be improved as well.
Answer: I have modified Figure 2
- Minor: the affiliation next to the authors’ name should be superscripted i.e.
Answer: Yes, I made the modifications. Thank you
Reviewer 2 Report
I would like to thank authors for clear and well presented study on computerized dynamic posturography of balance function characteristics in older patients with Orthostatic Hypotension.
I have two comments regarding manuscript:
1) The tittle of article emphasizes the quantitative assessment of balance functions but the assessment done by particular devices and six test conditions. Therefore I would suggest to consider to emphasize particular methodology used in assessment computerized dynamic posturography).
2) The study includes also laboratory tests which gives no statistically different results between the two study groups. Unfortunately, there was no clear explanation or reasoning described why laboratory tests were used for the study.
Author Response
On behalf of my co-authors, I would like to thank you for their positive and constructive comments and suggestions. Based on your modification suggestions, we make the following response.
1) The tittle of article emphasizes the quantitative assessment of balance functions but the assessment done by particular devices and six test conditions. Therefore,I would suggest to consider to emphasize particular methodology used in assessment computerized dynamic posturography).
Answer: Your comments were incredibly helpful, and as a result, I have included a description of CDP at the beginning of the third and fourth paragraphs of the discussion. It is hoped that this will highlight the significance of CDP in quantitative evaluation. The content of the conclusion has also been added accordingly. Thank you so much.
2) The study includes also laboratory tests which gives no statistically different results between the two study groups. Unfortunately, there was no clear explanation or reasoning described why laboratory tests were used for the study.
Answer: Thank you for your question. Hypertension, diabetes,and low levels of 25hydroxyvitamin D are also risk factors for OH. Therefore, in the second half of the fourth paragraph of the discussion, we analyze the laboratory indicators. As there is no statistical difference between the two groups, they are not the main factors influencing OH.